# Threats of Climate Change to Freshwater Ecosystems in Pakistan: eDNA Monitoring Will Be the Next-Generation Tool Used in Biodiversity, Conservation, and Management

**DOI:** 10.3390/biology14091191

**Published:** 2025-09-04

**Authors:** Ghazanfer Ali, Sidra Abbas, Satoshi Nagai, Norhafiza Mohd Arshad, Subha Bhassu

**Affiliations:** 1Animal Genetics and Genome Evolutionary Laboratory, Institute of Biological Sciences, Universiti Malaya, Kuala Lumpur 50300, Malaysia; s2125068@siswa.um.edu.my; 2Department of Zoology, University of Jhang, Jhang 35200, Pakistan; dr.sidraabbas@uog.edu.pk; 3Fisheries Technology Institute, Japan Fisheries Research and Education Agency, Yokohama 220-6115, Japan; snagai@affrc.go.jp; 4Center for Research in Biotechnology for Agriculture, Universiti Malaya, Kuala Lumpur 50300, Malaysia; norhafiza@um.edu.my

**Keywords:** climate change, freshwater ecosystem, invasive species, eDNA as biomonitoring tool, biodiversity conservation

## Abstract

**Simple Summary:**

The current rates of climate change are unprecedented, and biological responses to these changes are occurring rapidly at the ecosystem, community, and species levels. Climate change is a significant threat to freshwater biodiversity, resulting in higher extinction and local extinction rates among freshwater species than terrestrial taxa. In the current era of rapid environmental change and species loss, environmental DNA (eDNA) offers a more efficient and rapid method to monitor and conserve freshwater biodiversity. The use of eDNA metabarcoding has been recognized as a powerful technique for obtaining extensive data on biodiversity.

**Abstract:**

Freshwater ecosystems are a significant entity that govern the livelihood of people and are an important source of food, employment, and recreation. However, climate change is impacting freshwater ecosystems by altering their natural habitats. The purpose of this review is to highlight the vulnerability of freshwater fish to climate change. Climate change is invariably affecting natural ecosystems everywhere and in every part of the world, but these threats are more severe in Pakistan. Freshwater fish are important biotic drivers of freshwater ecosystems. Unfortunately, uncertain climate changes and anthropogenic activities have led to a decline in the diversity of these fishes. Rising temperatures, melting glaciers, changes in seasonal patterns, disturbances in the natural flow of rivers, pollution, and invasive species are major threats to native freshwater fish fauna, leading to a decline in fish diversity and population. *Tor putitora*, *Glyptothorax kashmirensis*, and *Triplophysa kashmirensis* are some of the species that are critically endangered in Pakistan due to these factors. In recent decades, insufficient attention has been paid to the freshwater ecosystem. This review of threats to the endemic fish species in this region is presented so that the government and policymakers can use this information as part of their management and conservation policy, thus safeguarding Pakistan’s fish industry. Environmental DNA (eDNA) biomonitoring is a new technique for assessing biodiversity and species distribution and can be useful for conserving biodiversity in this region. Another purpose of this review is to introduce this new conservation strategy to Pakistan.

## 1. Climate Change

Climate is the result of the interaction of weather components under different circumstances of time and space, including air temperature, pressure, solar radiation, precipitation, humidity, wind speed, etc., and alterations in these components are termed climate change [1,2]. Climate change is directly or indirectly associated with human activities that modify the composition of the overall atmosphere, in addition to natural climatic changes observed over comparable periods. Climate change has turned into a global environmental problem that has taken over the international agenda and is one of the biggest challenges for humanity [3]. The global average temperature has increased by 0.7 °C over the past century and is expected to continue to rise [4]. According to the predictions of the Intergovernmental Panel on Climate Change (IPCC), temperatures are anticipated to rise by 1.1 to 6.4 °C by the end of the 21st century compared to the 1980–1999 baselines [5]. It is predicted that average global precipitation has increased by 2% in the last 100 years and it is expected to increase in the future [5]. The impact of global climate change on hydro-ecological, freshwater, and agricultural systems are expected to vary across regions due to the influence of multiple environmental processes and human activities. [6,7]. In general, there is increasing evidence of climate change affecting ecosystems, including groundwater compatibility, surface water, and wetland ecosystem performance. However, its impact on water resources, including important hydrological metrics such as the magnitude, intensity, and frequency of rainfall, varies significantly at local and regional levels [8]. Globally, climate change is a significant threat that has already had profound impacts on natural ecosystems and biodiversity [4]. Evidence shows that increasing climatic change is responsible for geographic range shifts in species distributions, modifying phenology and migration patterns, diminishing the accessibility of suitable habitats for species, and disturbing important ecological interactions in communities [9,10]. Even though a small number of recent species extinctions have, as yet, been officially or tentatively attributed to climate change [11,12], data from the fossil record provide evidence that rapid climate change can be a fundamental driver of mass extinction, capable of eliminating up to 90% of all species [13,14,15,16]. This raises concerns about the adaptive potential of existing species to cope with future climatic changes [17,18]. The duration, frequency, and magnitude of extreme events have already increased in the last few years and are expected to worsen in the future with the intensification of anthropogenic activities [5]. There is sound scientific agreement that climatic change is not only factual but also has [19] and will continue to have continuous major impacts globally, including species extinction [20,21], seasonal changes in key ecological events, disturbances to the provision of natural goods and services, and even conflict in human societies [20]. These effects will expand to living things across all biological groups, including freshwater ecosystems. The effects of warming, and the consequent impacts on freshwater fish as a food source and environments providing employment, recreation, and tourism worldwide, [22] will be a tragedy.

## 2. Climate Change in Pakistan

It is very evident that climatic changes altering natural systems are happening everywhere in the world, but the alterations in climate extremes and their effect on Pakistan will be significant and have more drastic impacts [23]. These climate changes are anticipated to have adverse impacts on Pakistan [24] and are expected to have extensive and long-lasting effects on Pakistan’s environment and society [25]. Harsh weather conditions, drought, the melting of glaciers in the Himalayas, and increased heat in some parts of the country pose severe threats to many of the important rivers of Pakistan (Figure 1 and Figure 2). In terms of how extreme changes in climatic conditions will impact nations, Pakistan was placed fifth among the most affected countries between 1999 and 2018 [25]. Even though the contribution of Pakistan to global greenhouse gas (GhG) emissions is minimal, e.g., less than 1% of the world total [26], it is very susceptible to the adverse impacts of climate change happening in other parts of the world. Because Pakistan is poor in terms of technology and has poorer financial resources to acclimate to the harmful effects of climate change, its susceptibility has worsened [27]. Major threats faced by countries due to climate change include threats to food and water safety and the large migration of populations [28]. Pakistan’s economy is largely dependent on agriculture, which is the reason for its vulnerability to the circumstances caused by increasing climate change. Like other South Asian countries, Pakistan is facing a great threat because of climate change [29]. Extreme monsoons and cyclones are expected to increase in Pakistan as temperatures in the sea and the atmosphere rise [30]. Government forecasts highlight extensive developments in the intensity and frequency of life-threatening events, with irregular monsoon rains causing alternating floods and extreme droughts [27,29]. For instance, between 1998 and 2018, more than 150 life-threatening weather events were reported in Pakistan [31]. In 2022, catastrophic floods hit the country. The main causes of these were increased precipitation and glaciers melting, fueled by climate change. One-third of the country was underwater. According to Climate Minister Sherry Rehman, this “has exceeded every boundary, every norm we’ve seen in the past [32]” and 33 million people were affected [33] (Figure 3).

## 3. Drivers of Global Climate Change

Emissions of greenhouse gases have boosted the greenhouse effect, causing the surface temperature of the earth to increase. The burning of fossil fuels is causing the climate to change more rapidly than any other human activity [36,37]. In 2022, the global levels of major greenhouse gasses—carbon dioxide (CO_2_), methane (CH_4_), and nitrous oxide (N_2_O)—continued to increase, underscoring the persistent impact on climate change (Figure 4 and Table 1). CO_2_, the predominant greenhouse gas, was responsible for approximately 64% of the climate’s warming effect, with concentrations in 2022 surpassing pre-industrial levels by 50%, a pattern that extended into 2023. Almost all this increase is due to human activities [38]. The rise in CO_2_ from 2021 to 2022 was measured at 2.2 parts per million (ppm), a slight decrease from the previous year’s increase of 2.46 ppm and the average growth rate observed over the last decade. This decline may be attributed to the enhanced absorption of CO_2_ by natural carbon sinks, including terrestrial ecosystems and oceans, particularly following several La Niña years. Nonetheless, the anticipated onset of an El Niño event in 2023 could potentially disrupt this trend [39]. Methane, accounting for approximately 19% of the greenhouse warming effect, experienced a notable increase. While the growth from 2021 to 2022 was somewhat less than the unprecedented rate recorded between 2020 and 2021, it still exceeded the average growth rate observed over the past decade [39]. Human activities, particularly agriculture and fossil fuel extraction, are responsible for about 60% of methane emissions, with the remaining 40% stemming from natural sources [39,40]. Nitrous oxide is a significant greenhouse gas and a contributor to ozone depletion, representing approximately 7% of the radiative forcing attributed to long-lived greenhouse gasses [39]. The rise in N_2_O levels from 2021 to 2022 marked the largest annual increase ever documented, exceeding all prior growth rates. Emissions of N_2_O originate from natural sources (60%) and human activities (40%), including the application of fertilizers, biomass combustion, and various industrial operations [39,41]. Pakistan was identified by the United Nations Environment Program (UNEP) as one of the countries responsible for global greenhouse gas (GHG) emissions, contributing 1.02% to the world’s total [42,43]. In 2018, Pakistan emitted approximately 504.59 million tons of GHGs. The country is among the most affected by both the adverse impacts of climate change and deteriorating air quality due to pollution. GHG emissions are a primary driver of global warming, making them a critical focus of research in both the scientific and political arenas [42]. In 2019, the burning of fossil fuels, including coal, oil, and gas, released around 80 million tons of GHG, with carbon dioxide being the predominant component [26]. These emissions significantly contribute to air pollution and the decline in air quality. In response to these challenges, Prime Minister Imran Khan announced in 2020 that no further coal-fired power plants would be permitted in Pakistan [44].

## 4. Effects of Climate Change on Freshwater Ecosystems and Fishes

Perhaps, climate change has been affecting ecosystems for the last few years, and like many other unnoticed sources of environmental system degradation [45], may have altered our understanding of the ‘baseline’ conditions in freshwater ecosystems and important life history characteristics of freshwater fishes [46,47]. Over the last century, the global mean surface temperature has risen by 0.75 °C (possible range: 0.56–0.92), with the rate of increase accelerating since the 1970s [48]. This warming has also been documented in freshwater ecosystems worldwide [49,50,51]. Warming cannot be completely controlled, even if emissions are stopped or reduced, due to inertia in the climate system. Projections of the scale and rate of change differ between models, but by 2100, global atmospheric temperatures might rise 6.4 °C beyond existing levels [52] due to the abnormal weather conditions of the current biological times [53]. The scale of change is such that air temperatures once considered extremely high will rapidly become normal by mid-century [54]. In addition, unprecedented weather conditions will occur first in tropical areas and developing countries, where freshwater fishing is an important source of protein and income [55].

Freshwater ecosystems are considered sensitive indicators of climate change [56], with extinction representing the greatest risk to freshwater species resulting from its effects. [21]. There is scientific agreement that the increasing levels of greenhouse gas emissions have enhanced climatic change [20,57]; one effect of the past and present emissions of greenhouse gasses could be an increase in air temperature of around 6.4 °C by the end of the 21st century, even if firm mitigation policies are implemented. Many harmful effects have directly or indirectly impacted freshwater fish populations and their habitats [58,59]. The impacts of climate change are predicted to occur in temperate regions through increases in water temperatures, changes in rainfall systems, the beginning and extent of snow cover, changes in flow rates, and disturbance events, such as floods, wildfires, and pest infestations [59]. Since freshwater fishes are ectotherms, rising water temperatures might cause increased physiological stress, which in turn directly raises metabolic demand, affecting metabolism, productivity, reproduction, growth, and survival [60,61]. It was also found that changes in global rainfall patterns can affect the seasonal flow of rivers and their phenology [62]. Important life phases of some freshwater fishes, as well as their population dynamics, are dependent on the projected seasonal flow patterns [63]. Therefore, projected precipitation effects could lead to decreased reproductive success, alterations in species composition, and local extinction [64]. In freshwater ecosystems, certain indirect effects might occur through modifications in biotic processes, such as an increase in exotic species [65,66], a higher risk of parasites and diseases [67], and increased predation and competition [68]. Overall, this could alter food webs, fish communities, and, ultimately, ecosystems.

Various non-climatic anthropogenic stressors, such as overexploitation, water pollution, habitat changes and degradation, the generation of hydropower, flow modifications, and deforestation [69,70,71], are existing drivers of the freshwater fish decline worldwide [72]. The effects of these stressors on freshwater fish are expected to become more complex due to climate change [73]. In the current era of warming, terrestrial areas have warmed more rapidly than oceans, leading to an increase in water temperature in rivers, lakes, and wetlands globally [74]. This reflects a stronger relationship between water and air temperature in freshwater systems [75], which are already under pressure from a range of non-climatic stressors such as the over abstraction of water, eutrophication, habitat destruction, and the introduction of alien species [76]. Climate change has increasingly and continuously threatened the global environment [77], sustainable human development, and biodiversity, primarily by altering the water cycle and global thermal regimes and causing acidification [12,77].

Pakistan, a subtropical country, faces significant fluctuations in climate-related impacts due to its geographic and climatic positioning [78,79]. The country has experienced considerable climate change impacts in recent years [80], particularly its freshwater ecosystems, which support diverse fish populations. The heavy rainfall events and subsequent flooding in 2022 and 2023, exacerbated by climate change, resulted in the widespread disruption of freshwater habitats. Flooding significantly impacts the physical characteristics of rivers and streams, resulting in the erosion of riverbanks, alterations in river pathways, and the destruction of habitats essential for fish nesting and breeding [81,82,83]. These alterations can facilitate the introduction of new or invasive species into freshwater ecosystems, intensifying competition for resources and displacing indigenous species [82,84,85]. Additionally, the arrival of invasive species, along with the decline in water quality caused by increased sedimentation and pollution from floodwaters, can exacerbate the stress on native fish populations, adversely affecting their health and reproductive viability [86,87,88].

Pakistan’s freshwater ecosystems face not only these immediate challenges but also significant long-term threats from climate change. The disruption of hydrological cycles, particularly due to the accelerated glacial melting in the Himalayas and Hindu Kush, altered the flow dynamics of the Indus River, which depends heavily on glacial meltwater during dry periods [89]. As the temperature rises, glacial melting accelerates, resulting in temporarily increased river flows; however, this trend poses the long-term danger of a diminished water supply as glaciers continue to shrink [90]. Such alterations disrupt essential seasonal flow patterns that are vital for agriculture and ecosystem stability, while also increasing sedimentation in rivers, which negatively impacts water quality and the habitats of aquatic species [91].

Alterations in monsoon patterns have caused erratic rainfall, resulting in extended periods of drought as well as severe flooding [92,93]. This variability significantly impacts freshwater ecosystems, causing water-level changes that can devastate breeding sites for fish and other aquatic life [94]. While flooding may restore certain wetlands, it frequently introduces pollutants and sediments that compromise water quality [95,96], thereby affecting the health and diversity of aquatic habitats [97]. On the other hand, drought conditions diminish the water supply, leading to higher pollutant concentrations [98] and intensified competition among species for scarce resources [99]. For example, rivers like the Ravi and Sutlej have experienced reduced water flow and sections that have dried up, conditions worsened by rising temperatures and shifting precipitation patterns resulting from climate change.These factors, compounded by climate change, contribute to the decline in freshwater fish diversity, with some species becoming endangered or extinct in certain regions [100,101]. The complex interplay of these threats underlines the urgent need for integrated water management and climate adaptation strategies to protect Pakistan’s freshwater ecosystems and biodiversity.

The freshwater biodiversity of Pakistan, especially its fish species, is under significant threat due to climate change. The country is home to a wide variety of fish, including numerous endemic species that thrive in specific environmental conditions. As temperature increases and water flow patterns shift, these species encounter unprecedented challenges. An elevated water temperature can lead to lower levels of dissolved oxygen [102,103], putting stress on aquatic organisms [104,105] and potentially resulting in a shift in species dynamics, where more resilient and often invasive species may outcompete native ones [106]. Additionally, changes in river flow can interfere with migration patterns that are essential for the reproductive cycles of many fish species [107]. The overall effects of climate change on Pakistan’s freshwater ecosystems are significant, endangering biodiversity, water security, and the livelihoods of countless individuals. To tackle these issues, it is essential to implement thorough monitoring, adopt sustainable water management practices, and create adaptive strategies to lessen the adverse effects of climate change on these crucial ecosystems [108,109].

Fish biologists have been discussing climatic changes and their effects on freshwater fish for nearly four decades [110,111,112,113,114,115,116,117,118,119,120]. Three significant phenomena have been identified as widespread responses by fishes to the current warming of their habitats: (a) shifts in latitudinal distributions, with temperate and subtropical fishes moving poleward [121]; (b) alterations in the phenology of important life events, such as the timing of spawning, the length of reproductive periods, and migration [122,123]; and (c) a reduction in average body size, linked to a decline in the size of adult individuals, leading to an increased proportion of smaller and younger individuals in populations [124]. The underlying mechanisms of these phenomena are not fully understood; however, they are believed to be driven, at least in part, by the biological functions and responses of individual animals, reflecting evolutionary processes or phenotypic plasticity [125,126].

As freshwater ecosystems provide vital services to humanity and are highly threatened by climate change (Figure 5), further studies are necessary to enhance our understanding of how the climate impacts species within these systems. Once the significance of this issue is formally recognized, legislative organizations worldwide should prioritize climate change as a primary area of concern; additionally, international and national funding institutions are increasingly supporting research into climatic change. The Food and Agriculture Organization (FAO) of the United Nations has recognized the importance of climate change’s impacts on fisheries [127], and the IPCC assessment reports include references to freshwater systems [72,128]. Interested readers should refer to these resources for additional data and information.

## 5. Significance of Freshwater Ecosystems

Freshwater ecosystems play an important ecological role by providing economically essential products and services, and they provide vital habitats for a large number of reptiles, mammals, birds, fishes, and aquatic plants. They serve as habitats for various migratory and threatened species of fishes, reptiles, and birds. Additionally, freshwater ecosystems attract tourists by providing recreational areas for bird-watching, sports, and games. Freshwater ecosystems, particularly wetlands, play a key role in mitigating climate change through various ecosystem functions, including CO_2_ sequestration, shoreline stabilization, water purification, and flood control [130]. They support a wide range of ecosystem services that are threatened by climate change and invasive species, among other factors. Fortunately, there are many options for protecting and conserving freshwater ecosystems, which need to be addressed immediately [131] (Figure 5).

## 6. Key Threats to Freshwater Ecosystems

Globally, freshwater ecosystems and their associated biodiversity are at risk. Pollution, the introduction and spread of invasive species, habitat loss, overexploitation, the construction of instream barriers, hydrological modifications, excessive water abstraction, and the intensification of agricultural activities are the main threats to freshwater ecosystems and biodiversity [69]. Due to these compound risks and effects, the freshwater ecosystem is attracting continuous global attention in the quest to determine practical and adequate ways to combat the expected large-scale extinction of freshwater fish [132]. It is estimated that about 30% of freshwater fish species are at risk of extinction [133].

## 7. Freshwater Fishes

Fish are vital and diverse components of ecosystems, offering services that directly or indirectly improve human well-being. They are a rich source of proteins, vitamins, and minerals for human consumption [134]. Additionally, fish are also vital as they provide a means of employment and income to millions of people worldwide [134]. Freshwater fish are found on all continents except Antarctica. They are harvested from major rivers, lakes, wetlands, small ponds, and streams, making significant contributions to global fish production relative to their size [135]. Millions of people in developing countries rely on freshwater fisheries [55]. Freshwater fish differ from marine fish in terms of function, including differences in their trophic level, scale, and responses to natural and anthropogenic drivers [53,136]. To date, approximately 32,500 fish species have been described worldwide [137], with nearly 41% being freshwater species, 58% being marine species, and 1% being diadromous species [138]. Although freshwater makes up only about 0.3% of the Earth’s total water, it surprisingly hosts about 15,000 species of freshwater fish [139]. In Pakistan, 193 freshwater species have been reported [140]. The Indus River alone is home to more than 180 fish species [141].

Freshwater fisheries provide important ecosystem services to humanity [22] by offering food, income, and recreation through recreational, touristic, and commercial fisheries [137,142]. Each sector varies in its susceptibility to and ability to adapt to climate change. Freshwater fish make significant economic and social contributions worldwide [143]. According to an FAO report, 11.5 × 10^6^ tons of fish were captured from inland waters in 2011, marking an increase of more than 30 per cent since the early 2000s, with these catches increasing every year [144]. These statistics are potentially low and highlight the pressure placed on freshwater fisheries due to the world’s growing population. Globally, the supply of animal protein to humans is dominated by freshwater fishes, especially in Africa [142], where larger fisheries are found (estimated at 2,567,427 tons in 2010), mainly in the Great Lakes of Africa, as well as in productive rivers in Egypt and Nigeria, and the lakes, reservoirs, and wetland fisheries in the Sahel [145]. The FAO [146] observed that the recent developments in inland catches were linked with India, Bangladesh, China, and Myanmar and that more than 70 per cent of the world’s production comes from Asian countries. In 2022, global fisheries and aquaculture production reached a record 223.2 million tons, with 185.4 million tons coming from aquatic animals, and for the first time, aquaculture production surpassed capture fisheries. Asia dominated this production, contributing 167.1 million tons—75 per cent of the global total—and producing 70 per cent of the world’s aquatic animals [146]. Some commercially important freshwater fishes of Pakistan, along with their IUCN status, are listed in Table 2.

## 8. Freshwater Fishes Response to Climate Change

The most direct, fast, and common way in which fish respond to climate change is alterations in growth [147]. For example, the rapid somatic growth rates of fish are directly affected by temperature variations, including both air and water temperature [148,149]. Furthermore, climate change is predicted to alter the life history of plankton and benthos [150,151], which ultimately impacts the energy consumption and development of fish [152]. Alterations in fish growth are likely to have predictable long-term effects on population characteristics and features (e.g., egg size, age size structure, and reproductive phenology), recruitment, and dynamics (e.g., overwintering mortality) [153,154], and subsequently affect other biological processes at higher organization levels [151,155]. Therefore, understanding the fish growth responses to changing environmental conditions is a crucial step in predicting the impacts climate change will have on ecosystems and communities [156,157]. In response to the current climatic changes, freshwater fishes experience significant shifts and changes in their distribution, which seriously affect fish assemblage and composition [158]. About half of the species of freshwater fish are projected to become extinct in the coming decades, with a marked decrease in tropical areas and a higher likelihood of extinction for species with small body sizes and/or limited geographical ranges [159]. The response of species to climate change has been demonstrated globally in both terrestrial and aquatic environments. Changes in species composition, a reduction in geographic distribution, and alterations in ranges along gradients of altitude are important responses of species to climate change [160].

## 9. Biotic Components of Freshwater Ecosystems and Climate Change

The biotic component of freshwater ecosystems can also be altered by climate change, which has indirect effects on freshwater fish species. These effects can be produced by increasing the size of populations, the distribution of alien invasive species [66], changes in competition and predation rates [68], and an increase in pest occurrence and disease risk [67]. Exotic species or alien species, as part of the anthropogenic activity, also affect local aquatic systems. Exotic species can adversely impact local species and profoundly change aquatic ecosystems through alterations in trophic levels (competition, predation, changes in the food web), the introduction of diseases and parasites, modifications to fishes’ habitat, and spatial shifts (overcrowding and aggressive effects) [161]. The impact of alien gam fish, largely unpredictable in terms of time and space, and the establishment of relatively few species will lead to the extinction of many endemic species worldwide [162]. Along with other climatic factors, the blind introduction of alien species is one of the major problems decreasing the number of native fish species [161,163]. Worldwide, invasive species have been recognized as agents of local biodiversity loss. IUCN Pakistan [139] has also examined the risk factors for endemic fishes, with about 493 that are critically endangered, and provides recommendations to restore the habitats of endangered local fish species. The common carp is known as a pest due to its abundance and tendency to eliminate aquatic plants and reduce water clarity in the habitats of various aquatic species. In the United States, habitat destruction was a major issue affecting many fish species. However, the invasion of alien species was the main factor [164].

Climate change and the interaction between alien invasive and diseased species are concerns, and both are intensified by global trade [12]. Climate change is thought to be beneficial for invasive species. These changes speed up the rates of colonization via adaptive migration and undermine the integrity of in situ biotic assemblages. This allows species to reside in novel locations and climates, thus increasing the chances of colonization. If the invasive species is pathogenic, the likelihood of other species developing new diseases may increase [165]. Changes in climate and environmental conditions lead to changes in disease vectors (e.g., parasites and malaria mosquitos), which are likely released from natural control. Anthropogenic factors are responsible for the decline in commercially important fish populations in Pakistan. *Glyptothorax kashmirensis,* found in the Jhelum River drainage, has been declared Critically Endangered by the IUCN. Currently, the river Jhelum is blocked at many locations. This will alter this species’ habitat and consequently affect this fast-moving river species. Due to these severe, irreversible threats, a more than 80% decline is expected in the next five to ten years. *Tor putitora* has also been declared Critically Endangered in Pakistan by the IUCN due to severe population decline. This species is under pressure from habitat loss, overfishing, and alterations in habitat quality, which result in the demolishing of breeding grounds. Six species—*Ompok bimaculatus*, *Ompok pabda*, *Ailia coila*, *Wallago attu*, *Chitala chitala*, and *Bagarius bagarius*—are Near Threatened and one species, *Schizothorax plagiostomus,* is Vulnerable [166] (Table 3 and Table 4).

## 10. Climate Change and Abiotic Components of Ecosystem

Freshwater fish, as ectotherms, face greater physiological stress and metabolic requirements when exposed to high temperatures, which directly influence their survival, development, productivity, and reproduction [60,61]. Additionally, spatial and temporal changes to precipitation will alter climatic patterns, which will negatively impact life cycles, population dynamics, phenology, and reproductive successes [62,63]. In some cases, freshwater fishes have responded to the effects of climate change through modifications in their demographic processes, spatial distribution alterations, shifts in the timing of seasonal migration and breeding, and evolutionary changes [167]. The combined effects of climate change and other human stressors have affected the fishes’ ability to deal with or react to these alterations at a rate matching that of climate change [58,168]. Species are in danger when they have a restricted ability to adjust or react to rapid climatic changes [169]. These species have a higher chance of extinction and are typically the focus of adaptation efforts [168,169]. The unparalleled rate and severity of climate change poses a threat to the world’s freshwater fish population [20,170].

## 11. Climate Change and Hydrological Cycle

River networks are under the influence of climate change; many are also affected by existing stressors or have been exposed to a legacy of stressors. The general predictions regarding climate change’s effects on the river networks include changes in the discharge timing and intensity, an increase in extreme events of both peak and low flows, and an increase in river temperature [171]. The hydrological cycle is significantly affected by climate change in Pakistan’s rivers, and there have clearly been significant alterations in the flow systems of various rivers. These alterations in river and stream flow distress the productive capacity of freshwater ecosystems and aquatic biodiversity [8]. Among the freshwater reservoirs in Pakistan, several small and large tributaries join to form the Indus rivers system. The area occupied by Indus River system is about 1.12 million Km2, of which 47% is located in Pakistan, 39% in India, 8% in China, and only 6% in Afghanistan [136]. In the upper Indus basin, 13% of the mountains are covered by glaciers [172], and water flow in the Indus River is affected by the temperature, solar radiation, and precipitation [161]. In Pakistan, there is a monsoon season from July to September and 60 to 70 percent of the total precipitation takes place in this season, affecting the river flow [173]. The Indus River originates in Tibet from the Kailash range and is the backbone of the river systems in Pakistan and India. It flows westward and eventually falls into the Arabian Sea, draining an area of about 945,345 km^2^ and reaching a length of about 3180 km. The Indus River system comprises 27 tributaries and seven chief rivers: Zhob River, Sawat River, Kurraam River and Gomal River. The Ravi, Chenab, Jhelum, Sutlej, and Beas rivers flow from India to Pakistan and the Kabul River flows from Afghanistan toward Pakistan. Pakistan’s well known and extensive canal system covers an area of about 780,000 hectares, of which 9.7% is wetland: 27% is coastal wetland and 73% contains fresh water [174]. Good water quality and purity are vital for human survival, and their availability is shaped by both anthropogenic and human-induced factors. Unfortunately, water quality is declining rapidly day by day due to anthropogenic influences, industrialization, agricultural growth, and rapid urbanization [175]. Heavy metal pollutants can contaminate the environment for a long period and have irreparable impacts on the productive capacity of aquatic ecosystem [176]. Heavy metals, even at very low concentrations, are dangerous for freshwater and aquatic biodiversity [177]. The increasing intensity of pollution affects water quality and threatens the aquatic ecosystem’s integrity. The habitat of aquatic organisms has been destroyed in Pakistan due to transboundary management in the Satluj and Ravi. Over the past two decades in India, rapid economic growth and increasing agricultural, industrial, and urban development in the upper catchment areas have increased the level of toxic compounds in rivers [177].

## 12. eDNA Biomonitoring a Conservation Tool

Altering ecological communities in response to climate change and other anthropogenic activities has become a global issue [178]. A sharp decline in biodiversity has been one of the most critical challenges all over the world since the 20th century [179,180]. Due to these anthropogenic activities, increased rates of biodiversity loss, species extinction, and loss of ecosystem functioning have been observed worldwide [181], and the worldwide species extinction rate during this period exceeds that of prehuman periods [182]. This severely threatens the sustainability of ecosystems and human health [179,183]. Specifically, in aquatic ecosystems, a rapid decrease in fish diversity is one of the major problems in the management of fishery resources. This decrease is caused by overfishing, water pollution, habitat degradation, genetic pollution, climate change, and invasive species [184,185]. Therefore, it is necessary to take effective measures for fish protection. These measures are based on detailed information about fish distribution and population characteristics, as well as the physiological characteristics and ecological niche of different fish. However, due to the complexity of aquatic ecosystems and the diversity of fish migration routes [186], it is extremely difficult to accurately assess fish diversity. Traditional methods of investigating fish diversity, such as trawling, seining, electric fishing [180], underwater acoustics [187], and visual methods, may underestimate fish diversity, since some rare fish, e.g., endangered species and invasive taxa, are extremely difficult to detect [180].

In terms of the above-mentioned issues, environmental DNA (eDNA) provides an alternative means of detecting various types of fish. The use of eDNA, particularly metabarcoding, has gained attention as a powerful tool for assessing and monitoring aquatic biodiversity. By avoiding direct sampling, capture, and visual species observation, eDNA metabarcoding can reduce time and cost requirements, while supporting the conservation and management of ecosystems through improved species detection [188,189]. eDNA analysis using high-throughput sequencing has proven to be a very sensitive assay for the identification and assessment of single or multiple species in different samples [190]. The application of the eDNA method in freshwater ecosystems started in 2008, through the process of detecting the existence of American bullfrogs in a pond [191]. Since then, the eDNA method has increasingly been used for monitoring aquatic macro-organisms in both marine and freshwater ecosystems [192,193]. A big advantage of biodiversity assessment using eDNA metabarcoding is that it allows for effective and non-invasive species detection with little effort and minimum cost, even when conducting extensive surveys [194,195]. The eDNA method is defined as the sum of DNA fragments extracted directly from environmental components like soil, sediments, or water. The DNA fragments contain the intracellular genetic material released into water by biological cells, and the extracellular DNA released into water after cell structure lysis or death [196]. The eDNA method is based on amplicon and/or shotgun sequencing technologies. DNA fragments extracted from environmental components are identified using molecular biological means and the specific gene detection of target fish species. This can be used to analyze the distributional characteristics of target fish species. The eDNA method has progressed in terms of fish resource management, gathering information on endangered fish [197,198], investigating invasive species, and evaluating fish diversity [180,199] and fish prey diversity [200]. eDNA metabarcoding detected more fish species than those captured by traditional surveys like electrofishing [201] and underwater visual censuses [202]. In addition, eDNA metabarcoding has proven to be an efficient biomonitoring tool for surveying fish diversity, especially in regions that are often overlooked or difficult to access [203]. Studies have proven that eDNA metabarcoding can detect alien and endangered species that are not easily detected [188] (Figure 6). Traditional biomonitoring programs require vast efforts with repeated sampling over time and space. These methods are invasive because they use electrofishing and nets, which can destroy habitats. Moreover, they often fail to detect all species in an environment, and to identify species in collected samples, taxonomic experts are required. Due to these limitations, eDNA-based programs are an attractive approach for the conservation, management, and restoration of biodiversity [204,205].

The exploration of eDNA’s role in monitoring freshwater biodiversity can be greatly enhanced by examining its implications for Pakistan’s freshwater ecosystems, especially in light of the challenges presented by climate change. The country’s freshwater environments are increasingly vulnerable to severe weather phenomena, including intense rainfall and flooding, which disrupt these ecosystems and promote the spread of invasive species. These non-native species frequently outcompete indigenous fish populations, resulting in reduced biodiversity and a shift in the ecological equilibrium of these habitats. In this context, eDNA serves as a groundbreaking and effective method for assessing and preserving fish biodiversity, providing numerous benefits compared to conventional survey techniques [206,207]. A significant advantage of eDNA is its high sensitivity in detecting the presence of species, even when they are rare or difficult to find [207,208]. This holds particular significance in Pakistan, where extensive freshwater ecosystems are prevalent and challenging to monitor thoroughly through traditional approaches. For example, rivers such as the Ravi and Indus face considerable human-induced pressures, further intensified by habitat alterations driven by climate change [209,210]. eDNA addresses these challenges by enabling species detection at various spatial scales, eliminating the necessity for the physical capture or direct observation of organisms [211,212].

In Pakistan, eDNA metabarcoding could be utilized to evaluate fish biodiversity in key river systems like the Indus, Ravi, and Jhelum. These rivers are not only essential for maintaining biodiversity but also support the livelihoods of local communities [213]. By integrating eDNA techniques, researchers can generate detailed inventories of fish species, including those that are endangered or invasive, thus providing essential data for conservation and management efforts [214,215,216]. Moreover, eDNA serves as a valuable tool for tracking the proliferation of invasive species that pose risks to native fish populations, providing an early detection mechanism to inform prompt management strategies [217,218]. The effects of climate change in Pakistan, characterized by heightened flooding and alterations in habitats, highlight the necessity for advanced monitoring technologies such as environmental DNA (eDNA). Flood events can facilitate the introduction of new species, including invasive ones, into freshwater ecosystems, significantly impacting species diversity and ecological interactions [218,219]. eDNA serves as a vital tool for monitoring these swift ecological shifts by delivering immediate insights into species distribution and population levels [220,221,222,223]. This functionality is essential for effective biodiversity management amid environmental upheavals and for maintaining the resilience of freshwater ecosystems [224].

eDNA has the potential to transform ecological risk assessments, particularly regarding pollution and habitat degradation [225]. In vital aquatic ecosystems like the wetlands of Head Islam, eDNA can be utilized to evaluate the impact of pollutants on fish populations. While eDNA does not directly indicate pollution, its analysis provides a highly sensitive method for tracking biodiversity changes, serving as a valuable indicator of environmental health. This allows for the indirect estimation of the scope of pollution and its impact on ecosystems [226]. Such an approach would deepen our understanding of pollution’s influence on biodiversity and aid in formulating strategies to alleviate these effects. Overall, the application of eDNA in the monitoring of freshwater fish in Pakistan is not only feasible but also highly advantageous. By leveraging the strengths of eDNA, researchers can gain deeper insights into the status of fish populations, track changes over time, and inform conservation strategies. The adoption of eDNA in Pakistani freshwater ecosystems thus represents a forward-thinking approach that could significantly enhance efforts to conserve the country’s aquatic biodiversity.

## 13. Conclusions

Like other developing countries, Pakistan is extremely susceptible to the effects of climate change. Pakistan’s technological and financial limitations, along with its low GHS emission status, provide no protection from the negative effects of global climate change. As an agricultural country, consisting of arid and semi-arid regions and highly dependent on irrigated agriculture, Pakistan has been extremely threatened by climatic changes. Prolonged drought, higher levels of glacial melting, early and long summers, and short winters are some of the negative impacts of climate change, which also affect overall agricultural productivity and water sources in the country. The carbon sinks are declining rapidly because the forest cover is low in the country, at about 4.5%, with a high deforestation rate of about 0.2–0.4% per year. The long-lasting impacts of climate change are predicted to threaten our biodiversity, food security, water availability, human health, and overall well-being. Climate change in Pakistan seriously affects ecological fragility, freshwater ecosystems and fisheries, the agricultural sector, and human health. The increased warming will have consequences, leading to more rain, and could accelerate the hydrological cycles in most parts of Pakistan; this could lead to floods and drought in the region. In recent years, human activities and environmental changes have posed serious threats to freshwater habitats, resulting in an extensive decline in freshwater biodiversity. In the future, anthropogenic climate change and human pressure on freshwater ecosystems, including water diversion, abstraction, dam construction, and pollution, are predicted to have progressively important implications. Increases in the air temperature and alterations in precipitation patterns change water temperature and flow regimes globally, consequently affecting two important aspects of habitat for the survival of freshwater species. Fish are ectotherms, directly affected by water temperature, and the hydrological regime regulates the dynamics and structure of the freshwater habitat. Only a few studies have assessed the potential impacts of climate change on freshwater fishes, in contrast to many studies that examine the potential impacts of climate change on species in terrestrial systems. Climate change and other anthropogenic activities, such as pollution, dam construction, disturbances to the natural flow of rivers, and the introduction of invasive species, are responsible for the decline in local biodiversity. Many freshwater fishes are highly vulnerable, including *Tor putitora* and *Glyptothorax kashmirensis,* which are critically endangered in Pakistan. eDNA biomonitoring is a promising tool for the conservation and assessment of these species in the future. Environmental DNA (eDNA) refers to genetic material obtained directly from environmental samples, such as water, soil, or air, rather than from an individual organism. In the context of Pakistan’s freshwater ecosystems, eDNA can be a valuable tool for biodiversity conservation, especially as the country faces climate change, pollution, erratic monsoons, and floods. By enabling the non-invasive and comprehensive monitoring of aquatic species, eDNA helps to detect shifts in biodiversity, including the presence of invasive species and declines in native species. These real-time data can inform conservation strategies, guide habitat restoration efforts, and support adaptive management practices to mitigate the impacts of these environmental stressors.

## Figures and Tables

**Figure 1 biology-14-01191-f001:**
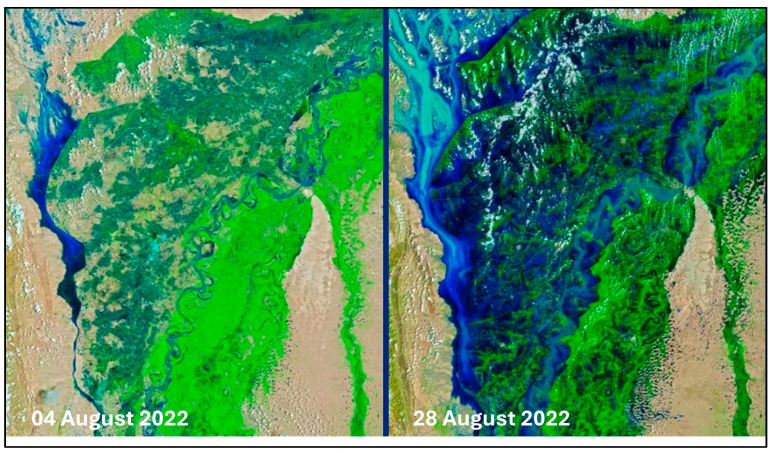
Satellite imagery showing a side-by-side comparison of southern Pakistan on 4 August 2022 and 28 August 2022. False-color satellite images from August 28th, showing floodwaters (blue) covering large portions of Pakistan’s normally arid, brown landscape (right). Infrared and visible light are combined to enhance the contrast between water and land. In the image of August 28th, how floodwaters dramatically altered the country’s landscape between early and late August is revealed. For instance, on August 4th, there was a stretch of approximately 30 miles (48 kilometers) of land separating the Indus River from Hamal Lake, situated west of the river in Pakistan’s Qambar Shahdadkot District (left). However, by August 28th, the floodwaters had caused the two bodies of water to merge (right). (Image credit: NASA Earth Observatory images by Joshua Stevens, using Landsat data from the U.S. Geological Survey and VIIRS data from NASA EOSDIS LANCE, GIBS/Worldview, and the Joint Polar Satellite System (JPSS). (https://earthobservatory.nasa.gov/images/150279/devastating-floods-in-pakistan, accessed on 31 August 2022).

**Figure 2 biology-14-01191-f002:**
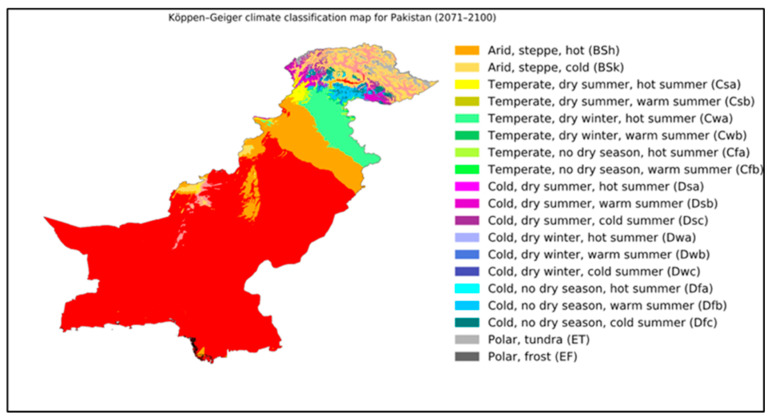
Future Koppen–Geiger climate classification map for Pakistan (2071–2100) [34]. A map for 2071–2100 under the most extreme climate change scenario, though mid-range scenarios are currently considered more likely [35].

**Figure 3 biology-14-01191-f003:**
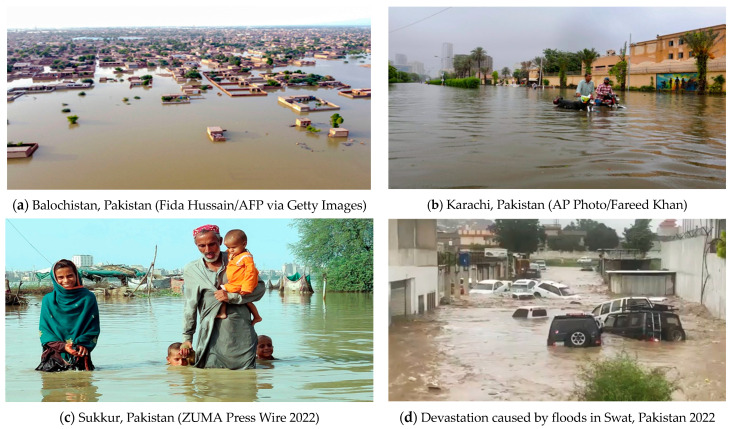
(**a**) Balochistan province on 29 August 2022. Image source: https://jordantimes.com/news/world/third-pakistan-under-water-flood-aid-efforts-gather-pace (accessed on 5 December 2023). (**b**) Pakistan’s largest city, Karachi, inundated with an entire summer’s worth of rain in one day, 25 July 2022. Image source: https://www.accuweather.com/en/severe-weather/pakistans-largest-city-inundated-with-an-entire-summers-worth-of-rain-in-one-day/1221899 (accessed on 5 December 2023). (**c**) A family navigates floodwater on Monday after a monsoon downpour in Sukkur, Pakistan. Image source: https://www.motherjones.com/environment/2022/08/pakistan-floods-flooding-monsoon-climate-change/ (accessed on 5 December 2023). (**d**) Screengrab of a video showing the devastation caused by floods in Swat, Pakistan 2022. Image source: https://www.geo.tv/latest/436128-emergency-declared-in-flood-ravaged-swat (accessed on 5 December 2023). (**e**) Flash flooding has devasted the lives of millions across Pakistan. Image source: https://www.axios.com/2022/09/15/pakistan-rains-increased-33-percent-climate-change (accessed on 5 December 2023). (**f**) Rescue workers help villagers evacuate from a flooded area after heavy rains fell in Lasbella, a district in Pakistan’s southwest Balochistan province, on 26 July 2022. Image source: https://www.theatlantic.com/photo/2022/08/photos-monsoon-flooding-in-pakistan/671278/ (accessed on 5 December 2023).

**Figure 4 biology-14-01191-f004:**
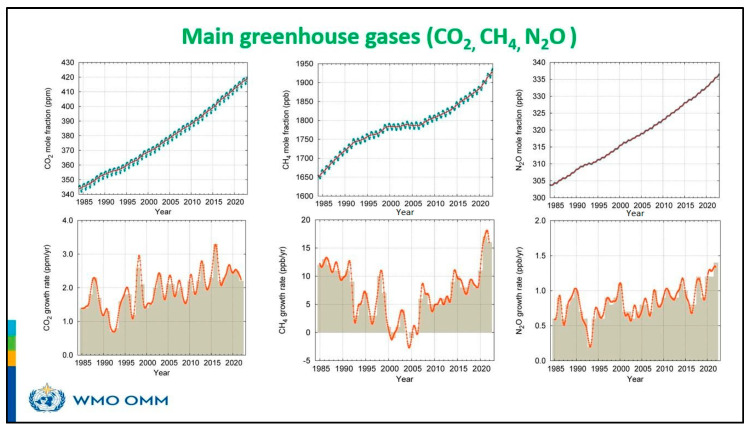
Graph showing the main greenhouse gas emissions for November 2023. Source: WMO 2024. (https://wmo.int/news/media-centre/greenhouse-gas-concentrations-hit-record-high-again. Accessed on 3 September 2024).

**Figure 5 biology-14-01191-f005:**
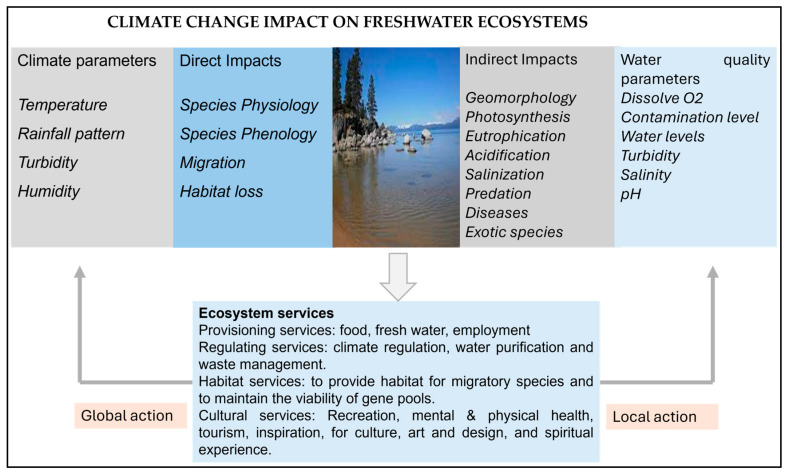
This Figure provides an overview of ecosystem services directly and indirectly impacted by climate change and local human activities. It illustrates the complex, cyclical relationship where the use of ecosystem services can, through direct and indirect mechanisms, influence the sustainability of those same services. Implementing global measures to mitigate harmful climate change impacts, along with local efforts to reduce disturbances in water quality, can help preserve these vital ecosystem services. Figure adapted from [129].

**Figure 6 biology-14-01191-f006:**
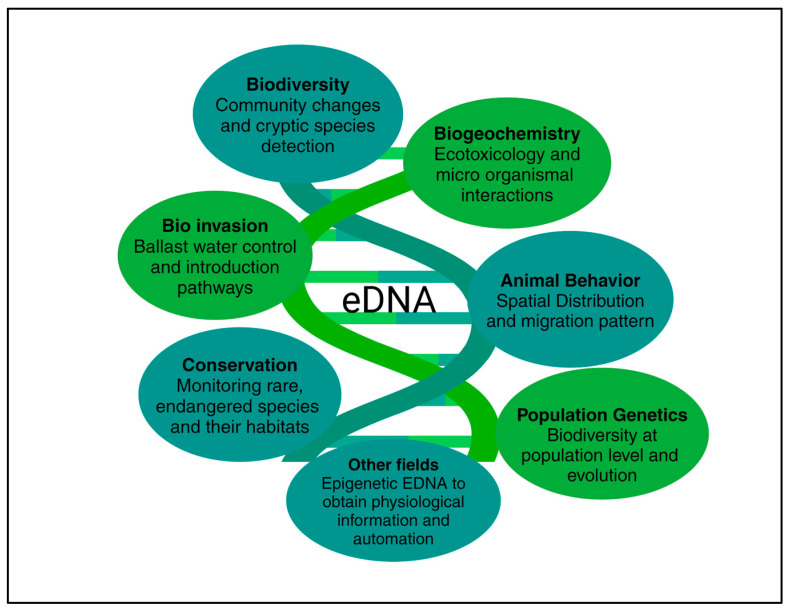
Environmental DNA and its services in biodiversity management and conservation.

**Table 1 biology-14-01191-t001:** Greenhouse gasses reach new levels.

	CO_2_	CH_4_	N_2_O
2022 global mean abundance	417.9 ± 0.2 ppm	1923 ± 2 ppb	335.8 ± 0.1 ppb
2022 abundance relative to 1750 ^a^	150%	264%	124%
2021–2022 absolute increase	2.2 ppm	16 ppb	1.4 ppb
2021–2022 relative increase	0.53%	0.84%	0.42%
Mean annual absolute increase over the past 10 years	2.46 ppm yr^−1^	10.2 ppb yr^−1^	1.05 ppb yr^−1^

^a^ Assuming a pre-industrial mole fraction of 278.3 ppm for CO_2_, 729.2 ppb for CH_4_, and 270.1 ppb for N_2_O. Source: WMO 2024. (https://wmo.int/news/media-centre/greenhouse-gas-concentrations-hit-record-high-again accessed 3 September 2024).

**Table 2 biology-14-01191-t002:** Some commercial freshwater fishes of Pakistan and their IUCN status.

Name of Fish	IUCN Status	Distributional Status	Commercial Value
Butter catfish (*Ompok bimaculatus*)	Near Threatened	Indigenous	NA
Pabdah catfish (*Ompok panda*)	Near Threatened	Indigenous	NA
Freshwater shark (*Wallago attu*)	Near Threatened	Indigenous	Very high
Gangetic ailia (*Ailia coila*)	Near Threatened	Indigenous	NA
Humped featherback (*Chitala chitala*)	Near Threatened	Indigenous	High
Gangetic goonch (*Bagarius bagarius*)	Near Threatened	Indigenous	High
Himalayan snowtrout (*Schizothorax plagiostomus*)	Vulnerable	Indigenous	High
Hilsa Shad (*Tenualosa ilisha*)	Not Evaluated	Indigenous	Very high
Mirgal (*Cirrhinus mrigala*)	Least Concern	Indigenous	Very high
Catla (*Gibelion catla*)	Least Concern	Indigenous	Very high
Thicklip labeo (*Labeo dyocheilus pakistanicus*)	Least Concern	Indigenous	High
Orangefin labeo (*Labeo calbasu*)	Least Concern	Indigenous	High
Kuria labeo (*Labeo gonius*)	Least Concern	Indigenous	High
Rahu (*Labeo rohita*)	Least Concern	Indigenous	Very high
Scaly Osman (*Diptychus maculatus*)	Not evaluated	Indigenous	High
Indus snowtrout (*Ptychobarbus conirostris*)	Not evaluated	Indigenous	High
Kunar snowtrout (*Racoma labiate*)	Vulnerable	Indigenous	High
Zig-zag eel (*Mastacembelus armatus*)	Least Concern	Indigenous	High
*Sperata seenghala*	Least Concern	Indigenous	Very high
Naziri bachcha (*Clupisoma naziri*)	Not evaluated	Indigenous	Very high
Rita Catfish (*Rita rita*)	Least Concern	Indigenous	Very high
Garua bachcha (*Clupisoma garua*)	Least Concern	Indigenous	Very high
Walking Catfish (*Clarias batrachus*)	Least Concern	Indigenous	High
Great snakehead (*Channa marulius*)	Least Concern	Indigenous	Very high
Chirruh snowtrout (*Schizopyge esocinus*)	Not evaluated	Indigenous	High
Ladakh snowtrout (*Schizopygopsis stoliczkai*)	Not evaluated/very rare	Indigenous	NA
Stoliczka triplophysaloach (*Triplophysa stoliczkai*)	Not evaluated/very rare	Indigenous	NA
Zebra Fish (*Danio rerio*)	Least Concern/very rare	Indigenous	NA
Whiptail Catfish (*Sisor rabdophorus*)	Not evaluated/very rare	Indigenous	NA
Bengala barb (*Megarasbora elonga*)	Least Concern/very rare	Indigenous	NA
Gangetic leaf fish (*Nandus nandus*)	Not evaluated/very rare	Indigenous	NA
Chameleon fish (*Badis badis*)	Not evaluated/very rare	Indigenous	NA
Gangetic mud eel (*Monopterus cuchia*)	Not evaluated/very rare	Indigenous	NA
One-stripe spiny eel (*Macrognathus aral*)	Vulnerable	Indigenous	NA
Punjab razorbellyminnow (*Salmophasia punjabensis*)	Not evaluated	Endemic	NA
Naseeri baril (*Barilius naseeri*)	Not evaluated	Endemic	NA
Pakistani baril (*Barilius Pakistanicus*)	Least Concern	Endemic	NA
Blue rahu (*Labeo caeruleus*)	Least Concern	Endemic	NA
Days’ labeo (*Labeo nigripinnis*)	Not evaluated	Endemic	NA
Balochistan labeo (*Labeo gedrosicus*)	Not evaluated	Endemic	NA
Macmahons’ labeo (*Labeo macmahoni*)	Not evaluated	Endemic	NA
Zhob mahasheer (*Naziritor zhobensis*)	Not evaluated	Endemic	NA
Punjab barb (*Puntius punjabensis*)	Least Concern	Endemic	NA
Salt Range barb (*Puntius waageni*)	Vulnerable	Endemic	NA
Javeds’ loach (*Botia javedi*)	Not evaluated	Endemic	NA
Swat loach (*Schistura alepidota*)	Not evaluated	Endemic	NA
Anambar loach (*Schistura anambarensis*)	Least Concern	Endemic	NA
Arifs’ loach (*Schistura arifi*)	Least Concern	Endemic	NA
Panjgur loach (*Schistura baluchiorum*)	Not evaluated	Endemic	NA
Kurram loach (*Schistura curtistigma*)	Vulnerable	Endemic	NA
Hangu loach (*Schistura fascimaculata*)	Not evaluated	Endemic	NA
Pishin loach (*Schistura kessleri*)	Not evaluated	Endemic	NA
Parachinar loach (*Schistura lepidocaulis*)	Not evaluated	Endemic	NA
Kohat loach (*Schistura kohatensis*)	Not evaluated	Endemic	NA
Mach loach (*Schistura machensis*)	Vulnerable	Endemic	NA
Khyber loach (*Schistura microlabra*)	Vulnerable	Endemic	NA
Rawlakot loach (*Schistura nalbanti*)	Not evaluated	Endemic	NA
Zhob loach (*Schistura pakistanica*)	Vulnerable	Endemic	NA
Pakhtunkhwa loach (*Schistura parashari*)	Near Threatened	Endemic	NA
Chenab loach (*Schistura shadiwalensis*)	Least Concern	Endemic	NA
Hazara Loach (*Triplophysa hazaraensis*)	Vulnerable	Endemic	NA
Nazir triplophysaloach (*Triplophysa naziri*)	Least Concern	Endemic	NA
Yasin triplophysaloach (*Triplophysa yasinensis*)	Least Concern	Endemic	NA
Horas’ mystus (*Mystus horai*)	Vulnerable	Endemic	NA
Pakistani gagata (*Gagata pakistanica*)	Not evaluated	Endemic	NA
Naziri catfish (*Glyptothorax naziri*)	Near Threatened	Endemic	NA
Kalabagh nangra (*Nangra robusta*)	Not evaluated	Endemic	NA
Sindh catfish (*Ompok Sindhensis*)	Not evaluated	Endemic	NA

NA: Commercial status not valuated; source: www.fishbase.se/search.php. Accessed on 22 November 2023; www.iucnredlist.org. Accessed on 5 December 2023.

**Table 3 biology-14-01191-t003:** Exotic fishes found in the river system of Pakistan.

Species	Occurrence	IUCN Status	Commercial Value
Goldfish (*Carassius auratus*)	Introduced	Least Concern	High
Grass carp (*Ctenopharyngodon idella*)	Introduced	Not evaluated	Very high
Common carp (*Cyprinus carpio*)	Introduced	Least Concern	Very high
Mosquito fish (*Gambusia affinis*)	Introduced	Least Concern	Minor commercial
Silver carp (*Hypophthalmichthys molitrix*)	Introduced	Not evaluated	Very high
Bighead carp (*Hypophthalmichthys nobilis*)	Introduced	Not evaluated	Very high
Rainbow trout (*Oncorhynchus mykiss*)	Introduced	Least Concern	Very high
Blue tilapia (*Oreochromis aureus*)	Introduced	Not evaluated	Very high
Mozambique tilapia (*Oreochromis mossambicus*)	Introduced	Vulnerable	Highly commercial
Nile tilapia (*Oreochromis niloticus*)	Introduced	Least Concern	Highly commercial
Sea trout (*Salmo trutta*)	Introduced	Least Concern	Commercial

**Table 4 biology-14-01191-t004:** Endangered and Critically Endangered species.

Common Name	Scientific Name	IUCN Status	Occurrence	Commercial Value
Kashmir catfish	*Glyptothorax kashmirensis*	Critically Endangered	Endemic	NA
Verinag triplophysaloach	*Triplophysa kashmirensis*	Critically Endangered	Endemic	NA
Golden mahasheer	*Tor putitora*	Endangered	Indigenous	Commercial
Wana garra	*Garra wanae*	Endangered	Endemic	NA
Havelian loach	*Schistura afasciata*	Endangered	Endemic	NA
Harnai loach	*Schistura harnaiensis*	Endangered	Endemic	NA
Dera loach	*Schistura macrolepis*	Endangered	Endemic	NA
Pakistans’ batasio	*Batasio pakistanicus*	Endangered	Endemic	NA
Punjab catfish	*Glyptothorax punjabensis*	Endangered	Endemic	NA
Bhed catfish	*Glyptothorax stocki*	Endangered	Endemic	NA
	*Glyptothorax punjabensis*	Endangered	Endemic	NA
Harnai käärtrull	*Schistura harnaiensis*	Endangered	Endemic	NA

## Data Availability

All data are available upon request, and we will be collaborating with the FishBase Consortium on depositing the data via the World Fish Centre.

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
