# Peer review of "Threats of Climate Change to Freshwater Ecosystems in Pakistan: eDNA Monitoring Will Be the Next-Generation Tool Used in Biodiversity, Conservation, and Management"

_biology, 2025, doi:10.3390/biology14091191_

Round 1
Reviewer 1 Report
Comments and Suggestions for Authors
In this paper, the author analyzes the influence of climate change on the freshwater ecosystem in Pakistan, and discusses the possibility of environmental DNA for freshwater biological monitoring and protection. The results provide a scientific basis for the protection of freshwater ecosystem in Pakistan. Moreover, the paper has some problems. I would suggest minor modification before acceptance.
1. While the authors provide a comprehensive overview of the effects of climate change on freshwater ecosystems, there is a notable lack of details specific to the impacts and consequences on Pakistan's freshwater systems. It is recommended that the authors expand this section to include more detailed information about the unique challenges faced by Pakistani freshwater systems.
2. Some of the cited literature appears to be outdated, such as the mention of total greenhouse gas (GHG) emissions in Pakistan at 408 million tons in 2015 (line 146) and a 2012 observation by the Food and Agriculture Organization (FAO) (line 285). To strengthen the relevance and accuracy of the study, it is advisable for the authors to incorporate more recent data and references where possible.
3. The authors aptly highlight the significant technical advantages of eDNA in biological monitoring and conservation. However, the discussion on the application of eDNA in freshwater biodiversity monitoring, particularly in relation to fish species in Pakistan, is relatively limited. It would be beneficial to supplement this section with research data focusing on eDNA applications in Pakistani freshwater ecosystems, specifically for fish monitoring. Furthermore, by combining the technical strengths of eDNA with the context of Pakistani freshwater resources, the authors could explore how this innovative tool can be utilized for the conservation and management of freshwater fish populations.
Author Response
Thank you very much for your thorough review of the manuscript. We have carefully considered your comments and made the necessary revisions, which are highlighted in the re-submitted files. Below, you will find detailed responses to each of your suggestions, outlining how the changes have been implemented. We appreciate your valuable feedback and hope the revisions meet your expectations.
|
Sr. No |
Comments |
Solution |
|
1 |
While the authors provide a comprehensive overview of the effects of climate change on freshwater ecosystems, there is a notable lack of details specific to the impacts and consequences on Pakistan's freshwater systems. It is recommended that the authors expand this section to include more detailed information about the unique challenges faced by Pakistani freshwater systems. |
Literature further expanded following the suggestions and comments. Line 236-288 |
|
2 |
Some of the cited literature appears to be outdated, such as the mention of total greenhouse gas (GHG) emissions in Pakistan at 408 million tons in 2015 (line 146) and a 2012 observation by the Food and Agriculture Organization (FAO) (line 285). To strengthen the relevance and accuracy of the study, it is advisable for the authors to incorporate more recent data and references where possible. |
This section is updated with the latest information found with references. Line 140-175 |
|
3 |
The authors aptly highlight the significant technical advantages of eDNA in biological monitoring and conservation. However, the discussion on the application of eDNA in freshwater biodiversity monitoring, particularly in relation to fish species in Pakistan, is relatively limited. It would be beneficial to supplement this section with research data focusing on eDNA applications in Pakistani freshwater ecosystems, specifically for fish monitoring. Furthermore, by combining the technical strengths of eDNA with the context of Pakistani freshwater resources, the authors could explore how this innovative tool can be utilized for the conservation and management of freshwater fish populations. |
The section further expanded specific to Pakistan. Line 574-621 |
Reviewer 2 Report
Comments and Suggestions for Authors
The structure and content of the manuscriot are ok. There are however, several grammatical errors the authors should pay attention to. I have made some suggestions on the manuscrit for your consideration

The quality of the language should be improved. It need moderate modification
Author Response
Thank you very much for your thorough review of the manuscript. We have carefully considered your comments and made the necessary revisions, which are highlighted in the re-submitted files. Below, you will find detailed responses to each of your suggestions, outlining how the changes have been implemented. We appreciate your valuable feedback and hope the revisions meet your expectations.
|
Sr. No |
Comments |
Solution |
|
1 |
The quality of the language should be improved. It needs moderate modification. |
The article has been revised to improve the English and correct grammatical errors, following the suggestions and comments. |
Reviewer 3 Report
Comments and Suggestions for Authors
Dear authors:
The manuscript is very good and complete, with many recent references, and supported with classic and former references that complement recent.
In section 9 "Biotic Components of Freshwater Ecosystems and Climate Change", I suggest include the role of freshwater invertebrates, and vegetation (algae, macrophytes) because there are many species that can be bioindicators of environmental conditions, there are many references that support it.
Figs 1-4, data source very old, I suggest include more recent information.
Fig. 5. Who is the author of this figure? ellaborate by authors?
Many success and blessings !!
Author Response
Thank you very much for your thorough review of the manuscript. We have carefully considered your comments and made the necessary revisions, which are highlighted in the re-submitted files. Below, you will find detailed responses to each of your suggestions, outlining how the changes have been implemented. We appreciate your valuable feedback and hope the revisions meet your expectations
|
Sr. No |
Comments |
Solution |
|
1 |
In section 9 "Biotic Components of Freshwater Ecosystems and Climate Change", We suggest include the role of freshwater invertebrates, and vegetation (algae, macrophytes) because there are many species that can be bioindicators of environmental conditions, there are many references that support it. |
Thank you for your valuable suggestion. While the role of freshwater invertebrates and vegetation as bioindicators is indeed significant and well-supported by literature, this study specifically focuses on freshwater fish. The primary aim is to examine the impacts of climate change on fish species, which is why the section does not include invertebrates and vegetation. Expanding to include these components would broaden the scope beyond the study's objectives. However, we appreciate your input, and this could be a valuable consideration for my future research. |
|
2 |
Figs 1-4, data source very old, we suggest include more recent information. |
1. Figure 1 has been updated to include the recent flood incident faced by the country (line 111-118). 2. Figure 2 cannot be changed as it contains images of the severe 2022 flood, which resulted in significant loss of life and economic damage. This is the recent flood the country faced in 2022. 3. Figure 3 shows a map for 2071–2100 under the most extreme climate change scenario, though mid-range scenarios are currently considered more likely (line 135-138). 4. Figure 4's data sources have been updated with the most recent information (176-179). |
|
3 |
Fig. 5. Who is the author of this figure? elaborate by authors? |
Authors information are updated (338-344). |